# Impact of Air Pollution on the Health of the Population in Parts of the Czech Republic [note 1]

**DOI:** 10.3390/ijerph17186454

**Published:** 2020-09-04

**Authors:** Radim J. Sram

**Affiliations:** Faculty of Health and Social Science, University of South Bohemia, 370 05 Ceske Budejovice, Czech Republic; radim.sram@iem.cas.cz; Tel.: +420-724-185-002; Fax: +420-241-062-785

**Keywords:** air pollution, SO_2_, PAHs, PM2.5, DNA adducts, pregnancy outcome, sperm abnormalities, neurobehavioral changes, mortality

## Abstract

Thirty years ago, Northern Bohemia in the Czech Republic was one of the most air polluted areas in Europe. After political changes, the Czech government put forward a research program to determine if air pollution is really affecting human health. This program, later called the “Teplice Program”, was initiated in collaboration with scientists from the United States Environmental Protection Agency (US EPA). This cooperation made possible the use of methods on the contemporary level. The very high concentrations of sulphur dioxide (SO_2_), particulate matter of 10 micrometers or less (PM10), and polycyclic aromatic hydrocarbons (PAHs) present in the air showed, for the first time, the impact of air pollutants on the health of the population in mining districts: adverse pregnancy outcomes, the impact of air pollution on sperm morphology, learning disabilities in children, and respiratory morbidity in preschool children. A surprising result came from the distribution of the sources of pollution: 70% of PM10 pollution came from local heating and not from power plants as expected. Thanks to this result, the Czech government supported changes in local heating from brown coal to natural gas. This change substantially decreased SO_2_ and PM10 pollution and affected mortality, especially cardiovascular mortality.

## 1. Introduction

Mining districts in Northern Bohemia, the northern region in the Czech Republic, were in the late 1980s one of the most air polluted regions in Europe. Northern Bohemia is a highly industrialized coal basin. Brown coal containing 1–5% sulphur was used for power plants, industry, and local heating. Geographically, this area is a valley sandwiched between the Ore Mountains approximately 1000 m above sea level to the north and the Middle Bohemia Highlands, approximately 800 m above sea level to the south. The geographic location and prevailing winds from the northwest and southwest give rise to frequent inversions. The average concentration of sulphur dioxide (SO_2_) in the years 1982–1990 was 103 µg/m^3^, and that of total suspended particles (TSP) was 102 µg/m^3^ [1]. Air pollution from these sources caused extensive deforestation of conifers in the Ore Mountains.

Air pollution significantly affected human health. Kotesovec et al. [2] observed that increased daily mortality was related to air pollution for total mortality, cancer, and cardiovascular mortality, and significant shortening of life expectancy by 2 years for males and females. Sram [3] studied the impact of air pollution on pregnancy outcomes, diagnosed as congenital anomalies (CGA) or lower birth weight (LBW) in medical records of maternity hospitals between the years 1982 and 1986. In the district of Usti nad Labem, 7644 pregnancies were diagnosed with 9.8% CGA. In the district of Teplice, 7190 pregnancies were diagnosed with 8.2% CGA. This was approximately 4–5 times higher in this region than in other parts of the Czech Republic, according to official records. Similarly, in the same districts, more children were born with birth weights lower than 2500 g (LBW, 7.5–9.2% vs. cca 4.5% nationwide). Morbidity of children in the mining districts of Northern Bohemia differed significantly from the morbidity nationwide as follows [4]:for children 0–6 years old:respiratory diseases: 2.90 vs. 0.54 nationwide (No. of cases/100),mental illness: 1.06 vs. 0.53 nationwide (No. of cases/100);for children 7–15 years old:respiratory diseases: 1.40 vs. 0.45 nationwide (No. of cases/100),mental illness: 4.09 vs.2.00 nationwide (No. of cases/100).


The health consequences of environmental pollution became one of the major concerns of the Czech government after political changes in 1989. At the end of 1990, the government put forward an interdisciplinary project later called the Teplice Program in order to analyze the impact of air pollution on human health in the mining districts [5]. The mining district of Teplice in Northern Bohemia was used as the polluted district for this program. The district of Prachatice in Southern Bohemia had some of the cleanest air in the Czech Republic and was used as the control district (Figure 1). The distance between those two districts is 240 km. The Teplice district had 127,500 inhabitants and an area of 469 km^2^, of which a large part had been devastated by the strip-mining of coal and associated industrialization. The district of Prachatice had 51,500 inhabitants and an area of 1375 km^2^, of which 52% was woodlands. In 1993, the average PM10 concentrations in Teplice were 76 vs. 38 µg/m^3^ in Prachatice. Similarly, PM2.5 concentrations were 64 vs. 32 µg/m^3^, respectively, and benzo[a]pyrene (B[a]P) 3.7 vs. 2.5 ng/m^3^ [6,7].

The Teplice Program was initiated by the Czech Ministry of Environment. The research program was prepared in collaboration with the United States Environmental Protection Agency (US EPA) and included air pollution monitoring, human exposure, biomarker studies, and health effects studies. Conclusions from the Teplice Program are also relevant today because there are many countries that still rely on burning coal for home heating.

## 2. Teplice Program

### 2.1. Air Quality Monitoring

Pinto et al. [8] collected aerosol samples from Teplice in February–March and May–July 1992, and from Teplice and Prachatice during three periods: January–March 1993, May–August 1993, and November 1993–March 1994. Ambient aerosol and acidic gas samples were collected by the versatile air pollution sampler (VAPS). The samples were analyzed for three indicators of air pollution: SO_2_, PM2.5 and B[a]P; not all samples were analyzed for all indicators. These findings are shown in Table 1.

Pinto et al. [9] later used the ambient monitoring and source characterization data to determine the relative contributions of different source categories to the level of ambient PM2.5 in Teplice and Prachatice (Figure 2).

Prior to this analysis, it was believed that the main source of PM2.5 pollution was from power plants. However, in the period with fewer and less severe inversions (January–February 1994), the most significant source was from local heating (31.2% + 25.9% + 13.5% = 70.6%) compared to 15.2% from power plants and 4.7% from mobile sources. What was originally postulated as incinerator activity was in fact waste burned in local heating sources [8].

The results of study of pollution sources were extremely significant: The high SO_2_ content measured in Teplice was undeniable and was tied to the use of brown coal in local heating. Therefore, at the end of 1994, the Czech government approved 6.2 billion CZK to convert local heating in all the mining districts from brown coal to natural gas. This change substantially decreased air pollution by lowering levels of SO_2_ as well as PM2.5.

### 2.2. Genotoxicity and Embryotoxicity of Urban Air Particulate Matter

PM10 were collected daily in Teplice and Prachatice using the HiVol air sampler Anderson equipped with Pallflex filters 20 × 20 cm (TA60A20) during winter (October–March) and summer (April–September) in the years 1993–1994. The organic mass of crude extracts was dissolved in dimethyl sulfoxide (DMSO). The in vitro acellular assay for DNA adducts was analyzed by ^32^P-postlabeling, while embryotoxicity assay was performed using the Chick Embryotoxicity Screening Test (CHEST) [10,11]. The characteristics of the winter air samples are as follows: Teplice PM10 69.3 µg/m^3^, B[a]P 7.42 ng/m^3^; Prachatice PM10 29.6 µg/m^3^, and B[a]P 5.37 ng/m^3^. ^32^P-postlabeling DNA adducts were analyzed via high-performance liquid chromatography (HPLC) to identify some of the major DNA adducts. Using CHEST, embryotoxicity was defined as the sum of dead and malformed embryos. Identified DNA adducts were derived from 9-OH/B[a]P, anti-BPDE, B[b]F, B[j]F, B[k]F, CHRY, B[a]A, and I[c,d]P. The radioactivity of these adduct spots accounted for approximately 50% of total radioactivity detected along the diagonal zone. A good correlation between DNA adduct levels formed in the presence of the S9 metabolic activation system and the dose inducing 50% of exposed embryos malformation/or death was observed (*r* = 0.773, *p* < 0.001). In assays from both Teplice and Prachatice, the highest activity was found for fractions containing mainly polycyclic aromatic hydrocarbons (PAHs). These results agreed with those of other studies [12,13] which show that PAHs account for most of the mutagenic activity present in the neutral fraction of urban air.

This was the first report comparing the biological activities of complex mixtures in short-term assays with remarkably different end points, such as DNA adducts formation and embryotoxicity.

The results indicate that PAHs are a major source of genotoxic activities of organic mixtures associated with urban air particles in both districts. The results also confirmed similarities of the major emission sources of organic compounds in both districts which are presumably residential home heating in the winter and motor vehicles in the summer.

### 2.3. DNA Adducts and Personal Air Monitoring

Binkova et al. [14] analyzed the effect of carcinogenic PAH (c-PAH) exposure on DNA adducts (DNA isolated from WBC (white blood cells) detected by ^32^P-postlabeling) in a group of 30 healthy women from the city of Teplice. The women worked outdoors as postal workers and gardeners. Personal samplers used for collecting respirable particles PM2.5 (<2.5 µm) were provided by the US EPA [15]. PM2.5 collected on quartz filters was extracted and analyzed by HPLC with fluorometric detection [15]. In the pilot study in November 1992, authors observed a significant difference in DNA adduct levels between smokers and nonsmokers. Sampling in nonsmokers was done on 24 and 26 November, when the concentration of c-PAHs was 14.9 ± 6.9 vs. 7.7 ± 3.3 ng/m^3^. In addition, the total DNA adduct levels differed significantly on those two days of sampling (5.59 ± 2.96 vs. 2.61 ± 1.40 adducts/10^8^ nucleotides, *p* < 0.05). Ten women nonsmokers participated in a follow-up study on four sampling days from October 1993 to February 1994 (Table 2). In both studies, there was a significant effect (*p* < 0.01) of sampling day on DNA adduct levels that was related to personal exposure data. Correlation analysis proved the relationship between c-PAH personal exposure and DNA adduct level (*r* = 0.621, *p* < 0.001). Therefore, the authors recommended using simultaneous personal exposure monitoring if WBC are to be used for DNA adduct analysis.

This paper is the first one to relate DNA adduct levels to c-PAH exposure determined by personal monitoring.

### 2.4. Pregnancy Outcome

This study evaluated the impact of air pollution and lifestyle variables on full-term singleton births of European origin in the Teplice district from April 1994 through March 1996 (N = 1943). A total of 190 (9.8%) infants were below the 10th percentile of birth weight for gestational age. Thirty-day averages for PM10 varied from 29 to 86 µg/m^3^, with a mean of 47.7 ± 12.6 µg/m^3^. Thirty-day averages for PM2.5 varied from 17 to 70 µg/m^3^, with a mean 35.7 ± 11.8 µg/m^3^. Elevated crude odds ratios (ORs) were observed for IUGR (intrauterine growth retardation) in the first month of pregnancy as follows: for PM10, medium 40 to < 50 µg/m^3^ (OR = 1.62, CI 1.02–2.50, *p* < 0.02); for PM2.5, high > 50 µg/m^3^ (OR = 2.64, CI 1.48–4.71, *p* < 0.001) levels. Results for PM10 and PM2.5 were similar, but only the adjusted OR for the high PM2.5 was statistically significant (OR = 1.68, CI 1.18–2.40, *p* < 0.05). Increases in IUGR during the first month of gestation were associated with PM10 concentrations over 40 µg/m^3^ and PM2.5 over 37 µg/m^3^ in the Teplice district (Table 3). These data suggest that exposure to particulate matter or associated air pollutants early in pregnancy may adversely affect fetal growth [16].

One possible explanation for this finding is that co-pollutants such as PAHs may interfere with fetal development as they are usually adsorbed on the surface of fine particles. Binkova et al. [11] observed that genotoxicity of particulate matter in the ambient air is related mainly to PAHs. Another study on the same population from Teplice and Prachatice suggested that DNA–PAH adducts in placentae were positively related to IUGR [17]. Both these studies indicate that PAHs are the major source of genotoxic and embryotoxic activities of organic mixtures associated with air pollution in the Teplice district.

Dejmek et al. [18] further analyzed single births in the Teplice and Prachatice districts in the study from April 1994 to March 1998. Concentrations of PM10 and PM2.5 were continually measured using VAPS. The carcinogenic PAHs (c-PAHs) were identified as chrysene, bez[a]anthracene, benzo[b]fluoranthene, benzo[k]fluoranthene, benzo[a]pyrene, dibenz[a,h]anthracene, and indeno[1,2,3-c,d] pyrene [15]. The pollutant data for each month were divided into low (L), medium (M), and high (H) concentrations, which were the same for PM10 and PM2.5 as described in Dejmek et al. [16]. Concentrations for c-PAHs were L = < 15 ng/m^3^, M = 15 to <30 ng/m^3^, H = 30 ng/m^3^ or higher. In the Teplice study, 3349 pregnancies were evaluated, and IUGR exhibited in 322 (9.6%) newborns. In Prachatice, 1505 pregnancies were evaluated, and IUGR exhibited in 124 (8.2%) newborns. IUGR was notably increased in the first month of gestation in Teplice, ORs for PM10 were M = 2.11 (CI, 1.03–2.02), H = 2.14 (CI, 1.42–3.23). Corresponding values for PM2.5 were M = 1.38 (CI, 0.95–1.92, *p* < 0.15) and H 1.96 (CI, 1.02–3.11, *p* < 0.002). In the district of Prachatice, a significant association was observed only for PM10: M = 2.11 (CI, 1.03–4.33). c-PAHs increased IUGR in Teplice in the first gestation month M = 1.59 (CI, 1.06–2.39, *p* < 0.025), H = 2.15 (CI, 1.27–3.63, *p* < 0.001) (Figure 3). The risk of an infant born with IUGR increases with the level of fine particles and c-PAHs in the first month of gestation. The association between PM10 and IUGR observed in a previous study by Dejmek et al. [16] may be explained in part by PAHs adsorbed to air particles.

The effects of PAHs on fetal development and growth may be explained by PAH penetration into the placenta and different fetal tissues [19,20,21,22,23] and by direct interference with placental growth factors [24,25].

According to Barker [26], reduced fetal growth is an important predictor of later adult health risks, such as noninsulin-dependent diabetes, hypertension, and coronary heart disease. Therefore, higher exposure to pollutants during the early stages of intrauterine life may be responsible for diseases in middle age.

Epstein et al. [27] studied the relationship between toxic trace metals and outcomes of first delivery in pregnant women in Teplice and Prachatice. Maternal and cord blood levels of lead, mercury, and cadmium were very low and arsenic was undetectable. No effect of these metals to low birth weight or IUGR was observed.

### 2.5. Biomarkers and Pregnancy, DNA Adducts

Topinka et al. [17] analyzed DNA adducts in human placenta related to air pollution in nonsmoking mothers from the districts of Teplice and Prachatice. Forty-nine placenta samples were from summer 1994 and forty-nine samples from winter 1994–1995 (each sampling N = 25 from Teplice, N = 24 from Prachatice). They observed 1.40 ± 0.87 and 1.04 ± 0.63 adducts per 10^8^ nucleotides for the Teplice and Prachatice districts, respectively. A significant difference between both districts in placental DNA adduct levels was found only for winter samples (1.49 vs. 0.96 adducts per 10^8^ nucleotides, *p* < 0.023). Positive glutathione S-transferase M1 (GSTM1) metabolic genotype was detected in 51 mothers, and GSTM1-null genotype was found in 47 subjects. Higher DNA adduct levels were detected in a group with GSTM1-null genotype (*p* < 0.01). This finding was more significant in the polluted Teplice district (*p* < 0.05).

In another study with 158 mothers (113 nonsmokers and 45 smokers), DNA adduct levels were significantly higher in the polluted region and in smoking mothers. Using multiple regression models to analyze the effect of c-PAH concentrations and vitamin C levels in nonsmoking mothers, an inverse relationship between vitamin C levels and DNA adduct levels was found (b = −0.513, *p* < 0.05). Higher DNA adduct levels were observed in nonsmoking mothers delivering children with IUGR (b = −0.741, *p* = 0.01) [28].

DNA adduct data in placenta related to the effect of c-PAH exposure are complementary with in vitro DNA binding activity and embryotoxicity studies [11]; this proved the genotoxic and embryotoxic potential of the organic extracts from the Teplice and Prachatice districts.

### 2.6. Semen Quality

Rubes et al. [29] also examined associations between exposure to episodes of air pollution and increased DNA fragmentation in human sperm in young men from Teplice (N = 36), who were sampled up to 7 times between the years 1995 and 1997. No significant associations were found between exposure to air pollution and routine semen testing, in terms of volume, concentration, total count, motile percentage, or percentage of normal morphology. Only sperm chromatin structure (SCSA) changes expressed as DNA fragmentation index (SCSA-%DFI) were significantly associated with exposure to high levels of air pollution as previously indicated by Selevan et al. [30]. In the comparison of air pollution in January 1996 vs. September 1997, PM10 was 52 vs. 25 µg/m^3^, and PAH was 145 vs. 30 ng/m^3^, which corresponded to SCSA-% DFI 20.3 (16.0–24.6) vs. 12.2 (9.5–14.8).

Rubes et al. [29] put forward the hypothesis that reactive metabolites of PAHs might reach the testicles and react with sperm DNA to form breaks, which cannot be repaired in epididymal sperm about 10 days before ejaculation. This may be manifested as increased SCSA-%DFI. The study also included measurement of blood lead, cadmium, and mercury, but these blood metals were not associated with air pollution.

These two studies from the Czech Republic [29,30] were the first epidemiological studies reporting associations between air pollution and altered semen quality as sperm chromatin structure.

Rubes et al. [31] later studied the impact of c-PAHs on sperm quality in city policemen in Prague by SCSA. Concentrations of B[a]P in February 2007 were 1.03 ± 0.77 ng/m^3^ while in May 2007 0.16 ± 0.05 ng/m^3^. Winter concentrations of B[a]P significantly increased sperm chromatid damage: hDFI was 7.31 ± 3.64% vs. 5.46 ± 3.21% in May. These data were later used for the evaluation of B[a]P health risk by the WHO (World Health Organization) in 2010: personal exposure to B[a]P over 1.0 ng/m^3^ predict DNA fragmentation in sperm [32].

### 2.7. Neurobehavioral Studies

The study of the impact of air pollution on a child’s neurodevelopment was started only in the districts of Northern Bohemia. The study analyzed symptoms of minimal brain dysfunction (MBD) in the 5080 children attending the second grade in the districts of Usti nad Labem, Teplice, and Jablonec nad Nisou. Behavioral changes were observed in children in the polluted districts where 4.8% attended special needs schools; for those children who attended the regular schools, 10% were diagnosed with MBD [33].

Therefore, the children living in these districts were at greater risk for learning disorders due to the significant levels of air pollution, especially the high concentration of SO_2_ in ambient air, compared to other children in the Czech Republic. According to the Czech Statistical Institute [4] in 1988, mental illness was diagnosed in 4.09% of children of the age group 7–15 years old in the mining districts vs. 2% in the Czech Republic. The effect of SO_2_ exposure during pregnancy in mice by Singh [34] demonstrated changes in behavior for the righting reflex and negative geotaxis as time progressed. Therefore, Sram [3] hypothesized that in utero exposure to environmental chemicals causes functional changes in the nervous system that are expressed as developmental disorders or other behavioral dysfunctions.

Otto et al. [35] assessed neurobehavioral functions using the Neurobehavioral Evaluation System (NES2, computerized assessment battery) [36] in 2nd-, 4th-, and 7th-grade students from Teplice and Prachatice (2nd-grade cohort N = 772, 4th-grade cohort N = 322, 7th-grade cohort N = 470 children) (Table 4).

Those results indicate a poorer performance on neurobehavioral tests and high prevalence of learning disabilities in children from the air-polluted mining district of Teplice [35]. Arsenic (As) and mercury (Hg) levels in hair and urine were low and were not associated with any performance measures. SO_2_ levels were markedly higher in the Teplice mining district. During the critical perinatal period (1982–1983) for the 7th-grade Teplice students, the mean ambient SO_2_ levels were 145.9 ± 24.6 µg/m^3^. In January–March 1993, the SO_2_ level was 153 µg/m^3^ in Teplice vs. 29 µg/m^3^ in Prachatice, while PM2.5 was 122 µg/m^3^ vs. 44 µg/m^3^. Results of Otto’s work [35] seem to correspond with the original idea about the negative impact of air pollution on neurobehavioral function in children.

Increased concentrations of PAHs in polluted air may affect neuropsychological development in children. Prenatal exposure to PAHs was studied in cohorts of children from New York (USA) [37,38,39,40], Krakow (Poland) [41,42] and Tongliang (China) [43]. All studies observed decreases of cognitive function, intelligence quotient, and decrease of white matter volume in the left hemisphere. Therefore, it may be hypothesized that changes in neurobehavioral function in the population in Northern Bohemia may be a consequence of the high concentration of PAHs present in the mining districts in the decades prior to the Teplice Program.

### 2.8. Mortality

Life expectancy in the district of Teplice in 1988 was 64.9 years for males and 73.9 years for females compared to 68.2 years and 75.4 years, respectively, in the Czech Republic [2]. Now, life expectancy in the Czech Republic is 76 years for males and 82 years for females, but in the mining districts, it is 2 years shorter for males as well as females [44]. The extension of life expectancy after political changes in 1989 was related to improvements in the health care system and the shift in lifestyles and disease awareness [45]. Modern health policies included screening programs for certain malignant neoplasms, tobacco/smoking bans in public places and atialcohol measures [46]. Declines in cardiovascular mortality were associated with improvements in prevention and/or treatment of ischemic heart disease [47].

Kotesovec and Skorkovsky [48] analyzed mortality in the mining districts (Chomutov, Most, Teplice, Usti nad Labem, and Decin) which had a total of 620,000 inhabitants in the period of high (1982–1994) and decreased pollution (1995–2004). The concentration of SO_2_ in those two periods was 93.09 vs. 22.73 µg/m^3^, and PM10 was 97.00 vs. 44.14 µg/m^3^. The total standardized mortality in the period of high pollution was 14.80/1000 for males and 13.60/1000 for females; cardiovascular mortality 7.45/1000 for males and 8.17/1000 for females; and respiratory mortality 0.85/1000 for males and 0.52/1000 for females. Total standardized mortality in the period of decreased pollution was 13.10/1000 for males and 12.25/1000 for females; cardiovascular mortality 6.33/1000 for males and 7.18/1000 for females; and respiratory mortality 0.56/1000 for males and 0.42/1000 for females. Comparing those two periods, total mortality decreased by 5.1% (4.2, 6.1) for males and by 2.7% (1.9, 3.6) for females; cardiovascular mortality decreased by 6.9% (5.6, 8.3) for males and by 3.7% (2.6, 4.9) for females; and respiratory mortality decreased by 17.8% (13.8, 21.9) for males and by 5.5% (1.2, 9.9) for females. The most significant decrease was observed with cardiovascular mortality in subjects under 60 years old: for males by 11.3% (8.5, 14.2) and for females by 12.2% (7.6, 17.1).

The substantial decrease of air pollution in the mining districts between the years 1995 and 2004 significantly decreased total, cardiovascular, and respiratory morbidity in both males and females. Kotesovec and Skorkovsky [48] calculated that within that period, 195 fewer males and 92 fewer females were dying each year, totaling 1950 males and 920 females living longer. These results prove the significant implication that decrease in air pollution has on the health of the people in the mining districts.

## 3. Discussion

Thirty years of research in the Teplice Program proved the impact of air pollution on the health of the affected population.

A negative impact of air pollution on the health of the population, especially children, was expected according to empirical knowledge of medical doctors in mining districts, including many pediatricians. Results from the Teplice Program not only confirmed this effect but further proved genetic damage in the affected population, including the effect on spermiogenesis and pregnancy outcomes as the newborns matured with decreased resistance to environmental factors.

All these negative outcomes of air pollution are compounded by smoking exposure, either by direct smoking or by second-hand smoke [49]. Results of neurobehavioral studies indicate affected neuropsychological functions, with a high prevalence of learning disabilities and behavior issues, in the children of smoke-exposed parents. It may be hypothesized that smoke exposure may be one of the causes of the multifactorial conditionality of ADHD (attention deficit and hyperactivity disorder). This is the reaction of an organism to injury of its parent’s health leading to genetic damage and subsequent injury, such as fetus weakening by lower birth weight, “immaturity” with subsequent defect in concentration and therefore learning challenges, hyperactivity, growing to psychasthenia, decreased frustration tolerance, and decrease of adaptation ability.

The Program results imply that PAH exposure in polluted air is a significant health problem for the Czech Republic: 62% of population is exposed to B[a]P concentrations higher than the EU standard of 1 ng/m^3^.

The data also indicate that the population in the mining districts was significantly affected by high air pollution between the years 1972 and 1994. This is probably the main reason life expectancy in mining districts is already 2 years shorter for males and females compared to nationwide expectancy. Pregnancy outcome studies observed significant genetic damage due to the increase of DNA adducts in newborns, as well as intrauterine growth retardation and lower birth weight.

During the period of high pollution in mining districts, there were approximately twice as many newborns with a lower birth weight. According to Barker [26], significantly increased cardiovascular morbidity and diabetes may be expected when these subjects reach middle age (approximately 50 years old). A simple calculation estimates that we may begin to see these increases starting this current year, 2020. This knowledge should be used to prepare a preventive program for residents 50 years of age and older in the mining districts to be checked by general practitioners for the first signs of cardiovascular morbidity and diabetes.

The scientists who participated in this research were satisfied not only with their scientific results, but also that their results were used to implement change, especially in supporting the decrease of air pollution by changing local heating sources to use natural gas instead of coal, as well as the reduction of industrial pollution.

The results from the Teplice Program pointed the way to study the effect of air pollution in Northern Moravia, the Moravian–Silesian Region (MSR), due to heavy industry (steel and coke production). The MSR population is exposed to B[a]P concentrations several times higher than the EU standard of 1 ng/m^3^/year. In the district Ostrava Radvanice–Bartovice, the concentration is B[a]P 7–10 ng/m^3^/year [7].

For the first time, whole genome microarrays were used to analyze the relationship between air pollution and bronchial asthma in children. The results indicate the distinct phenotype of asthma in children living in the polluted Ostrava region (non/allergic type) compared to children living in Prachatice (allergic type) [50]. In those children, DNA methylation was also studied, which significantly differed between Ostrava and Prachatice [51].

The impact of air pollution on newborns was studied in the exposed district of Karvina (MSR) and the control district of Ceske Budejovice (Southern Bohemia). The concentration of B[a]P in winter 2014 was higher in Karvina (5.36 ± 3.64 vs. 1.45 ± 1.19 ng/m^3^, *p* < 0.001). It significantly increased in newborns in Karvina. DNA adducts [52], oxidative damage (DNA oxidation, lipid peroxidation) [53], and whole genome expression revealed a deregulation of processes associated with immune response or oxidative stress response [54]. In the urine of newborns, monohydroxylated metabolites of PAHs (OH-PAHs) were analyzed. ΣOH-PAHs in newborns’ urine samples from Karvina were in the winter of 2013 3.3 times higher when compared with those of newborns from Ceske Budejovice [55].

All these studies imply the significant health risk of PAHs in polluted air.

Results obtained during the Teplice Program also showed new topics for research, as well as potential actions to implement. It can be predicted that there are other populations exposed to similar air pollution with similar health injury. Therefore, results already obtained in the Teplice Program may encourage scientists in other countries to study the contemporary impact of air pollution on a broad spectrum of possible health injury, especially in children.

## 4. Conclusions

In evaluating the Teplice Program after more than 20 years, it may be said that it was truly a unique international project. In collaborating closely with the US EPA, Czech scientists were trained in many new methodologies used in health studies. The results were unexpected in some cases and in other cases affirmed what various professionals had already suspected. The totality of the results is no less profound and is summarized below.

(1)In analyzing the sources of air pollution, approximately 70% of PM2.5 fine particles were attributed to local heating sources that used brown coal containing a high content of SO_2_. This result prompted the Czech government in 1994 to support the change of local heating in the mining districts from using coal to natural gas. This substantially decreased the concentration of SO_2_ and PM2.5 in the region;(2)In vitro studies, evaluated in terms of the level of DNA adducts, proved that PM10 extracts contained many c-PAHs and that those c-PAHs contributed to 50% of the genotoxicity of PM10 in the region;(3)The use of personal monitors and determining the level of DNA adducts in WBC of exposed subjects showed for the first time the relationship between c-PAH exposure and the impact on DNA adducts;(4)In the pregnancy outcome study, analyzing several thousand pregnancies over 4 years showed that the first month of gestation is the most sensitive to IUGR induction. The prevalence of IUGR was shown to be related to the concentration of c-PAHs > 15 ng/m^3^ (B[a]P > 2.8 ng/m^3^) adsorbed on PM2.5. This reduced fetal growth may substantially affect later adult health;(5)For the first time, the impact of air pollution on DNA fragmentation in sperm was demonstrated in the semen study;(6)In the neurobehavioral studies, poorer performance on neurobehavioral tests and high prevalence of learning disabilities in children from the polluted district was observed. The studies postulated a significant impact of air pollution that affects neurobehavioral function in children;(7)The mortality studies observed a significant decrease of life expectancy, approximately 2 years for males and females. Decreased air pollution later significantly decreased cardiovascular and respiratory mortality in mining districts;(8)It may be suspected that the health of the population in the mining districts of Northern Bohemia is significantly affected by decades of air pollution. High concentrations of c-PAH induced genetic damage as well as affected birth weight, which later exhibits as functional changes that increase morbidity. This damage seems to be long-lasting and can extend through an entire lifetime.

These facts should be used to prepare a new “Teplice Program” to learn the present quality of life in the mining districts and use new knowledge to prepare a preventive program to improve the population’s health in this region.

## Figures and Tables

**Figure 1 ijerph-17-06454-f001:**
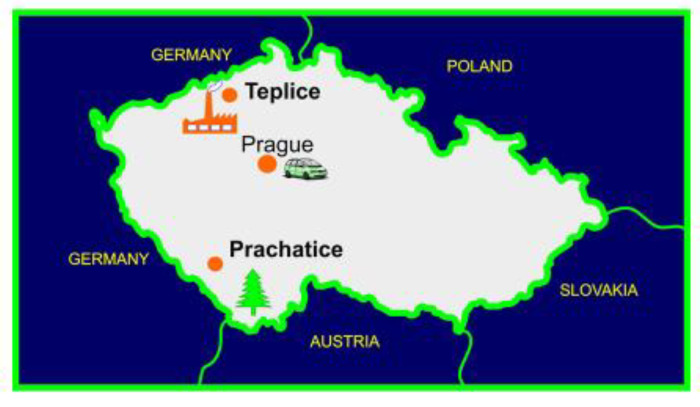
Map of the Czech Republic with the locations of studied districts.

**Figure 2 ijerph-17-06454-f002:**
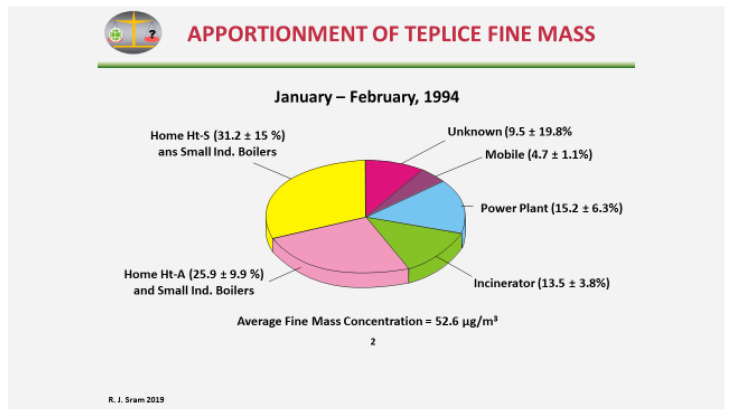
Sources of PM2.5 air pollution in the Teplice district.

**Figure 3 ijerph-17-06454-f003:**
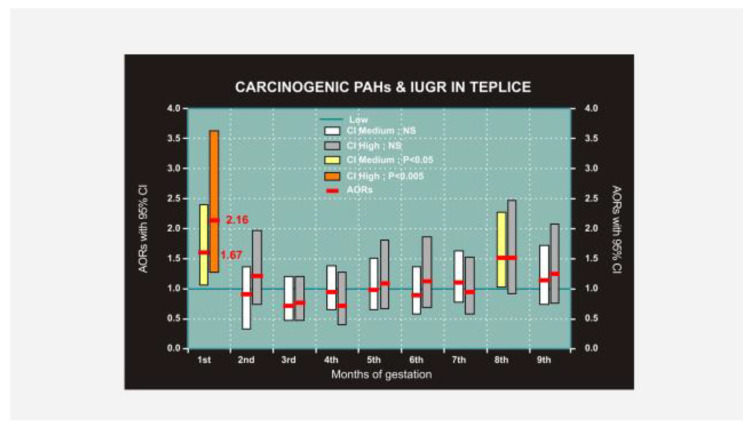
Impact of carcinogenic PAH (c-PAH) exposure to mothers in the Teplice district on IUGR in their newborns.

**Table 1 ijerph-17-06454-t001:** Concentrations of indicators of air pollution in Teplice (target population) and Prachatice (control group).

Indicators of Air Pollution	February–March 1992	May–July 1992	January–March 1993		May–August 1993		November 1993–March 1994	
	Teplice	Teplice	Teplice	Prachatice	Teplice	Prachatice	Teplice	Prachatice
SO_2_, µg/m^3^	135 ± 20	31.1 ± 4.7	153 ± 23	29.0 ± 4.4		4.4 ± 0.7		
PM2.5, µg/m^3^	68.0 ± 1.9	36.5 ± 1.2	122 ± 3.1	44.0 ± 0.8	28.7 ± 1.2	17.9 ± 0.4	51.1 ± 2.8	
B[a]P, ng/m^3^			8.0 ± 0.4	4.7 ± 2.4	0.5 ± 0.4	0.1 ± 0.05	5.5 ± 0.3	3.4 ± 0.5

**Table 2 ijerph-17-06454-t002:** DNA adducts in healthy women in Teplice.

Sampling					
	**November 1992**	**October 1993**	**November 1993**	**January 1994**	**February 1994**
PM2.5 µg/m^3^	53.5 ± 30.5	52.8 ± 39.4	106 ± 49.9	33.3 ± 14.1	39.3 ± 46.8
c-PAHs ng/m^3^	12.2 ± 5.6	14.5 ± 6.4	42.2 ± 19.9	21.3 ± 18.5	15.1 ± 6.0
B[a]P ng/m^3^	3.0 ± 1.3	2.8 ± 12.6	7.5 ± 3.6 *	3.8 ± 4.0	2.0 ± 1.1
DNA adducts/10^8^ nucleotides	5.73 ± 0.90	4.64 ± 1.95	6.81 ± 1.81	4.37 ± 2.05	3.96 ± 0.80

* *p* < 0.05 compared to samplings in October 1993, January 1994, February 1994.

**Table 3 ijerph-17-06454-t003:** Adjusted * odds ratio (AOR) of intrauterine growth retardation (IUGR) for PM10 by month of gestation.

Month	PM10: 40 to <50 µg/m^3^			PM10 > 50 µg/m^3^		
	AOR	CI	*p*-Value	AOR	CI	*p*-Value
1	**1.62**	**(1.07–2.50)**	**0.02**	**2.64**	**(1.48–4.71)**	**0.001**
2	1.09	(0.72–1.63)	0.69	1.01	(0.60–1.69)	0.98
3	1.02	(0.68–1.54)	0.93	0.87	(0.51–1.47)	0.59
4	1.27	(0.85–1.90)	0.25	0.93	(0.55–1.58)	0.78
5	0.92	(0.62–1.36)	0.66	0.82	(0.48–1.39)	0.46
6	0.95	(0.65–1.39)	0.77	0.74	(0.42–1.30)	0.29
7	0.83	(0.57–1.21)	0.33	0.83	(0.49–1.42)	0.50
8	1.22	(0.83–1.79)	0.31	1.16	(0.66–2.03)	0.61
9	1.03	(0.70–1.52)	0.88	1.25	(0.73–2.12)	0.42

* Adjusted for maternal height, pre-pregnancy weight, completed high school, currently married, month-specific smoking habits, year and season.

**Table 4 ijerph-17-06454-t004:** Percent of children referred for assessment of learning disabilities or behavioral problems.

Cohort	Teplice	Prachatice
2nd grade	26.6	12.9
4th grade	27.3	13.0
7th grade	25.6	13.1

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
