# Peer review of "Impact of Air Pollution on the Health of the Population in Parts of the Czech Republic [Author-notes fn1-ijerph-17-06454]"

_ijerph, 2020, doi:10.3390/ijerph17186454_

Round 1
Reviewer 1 Report
Please explain this statment, why do you think this?
The geographic location and prevailing winds from the northwest and
southwest give rise to frequent inversions.
Prior to this analysis, it was believed that the main source of PM2.5 pollution was from power plants. However, in the period without inversion (January – February 1994),...
I do not think, that during the mentioned time period ther was no inversion. Please think it over.
Author Response
Thank you for your comments.
l. 93 was corrected:
...in the period with fewer and less severe inversions...
Reviewer 2 Report
The problem of air pollution has long been not only the problem of scientists, but above all those who suffer from health effects. When it concerns children or pregnant women, it seems that the more it should be thoroughly analyzed. That is why I believe that the topic raised by the author and analysis of the research on the impact of air pollution on health, including the most vulnerable units, which have been carried out over the last 20 years, constitute an important contribution to this area of research.
Actually, the only aspect of the work that needs proofreading are minor editorial errors:
- incorrect record of sulfur dioxide (without subscript),
- in verse 34 the concentration unit was written incorrectly ("ug"),
- it would be necessary to standardize the record of other data, including spaces or no spaces next to the characters "=", "<", ">" and "%" - of course, this does not affect the substantive value of the work, which is indisputable, but at least clearly distract the addressee.
Author Response
Thank you for your comments.
Actually, the only aspect of the work that needs proofreading are minor editorial errors:
- incorrect record of sulfur dioxide (without subscript),
it was corrected
- in verse 34 the concentration unit was written incorrectly ("ug"),
it was corrected
- it would be necessary to standardize the record of other data, including spaces or no spaces next to the characters "=", "<", ">" and "%" - of course, this does not affect the substantive value of the work, which is indisputable, but at least clearly distract the addressee.
it was corrected
Reviewer 3 Report
This paper is a review of data collected in the “Teplice Program” and shows important results of the research conducted in the region of the Czech Republic in the 1980s and 1990s. In this study exposure to airborne pollution, several biomarkers and health outcomes were monitored over 20 years in Teplice, a region of the Czech Republic with high pollution levels, and compared with Prachatice, used as the control district because of having one of the cleanest air in the Czech Republic. The work is well written, logically and thematically coherent and the topic is of great scientific interest.
However, there are some points that can be improved (minor revision):
- At the end of the "Introduction" it would be useful to define better and more clearly what the aim of this revision is;
- In the "Introduction" the Author should indicate the distance between Teplice and Prachatice;
- In the "Discussion" the Author should discuss more in depth the results by comparing with those of other international and recent studies. [i.e. consider: “Air pollution and public health: A PRISMA-compliant systematic review” (https://doi.org/10.3390/atmos8100183); “UK air pollution and public health” (https://doi.org/10.1016/s0140-6736(17)31271-0)]
Author Response
Thank you for your comments.
At the end of the "Introduction" it would be useful to define better and more clearly what the aim of this revision is;
added
- In the "Introduction" the Author should indicate the distance between Teplice and Prachatice;
added - 240 km
- In the "Discussion" the Author should discuss more in depth the results by comparing with those of other international and recent studies. [i.e. consider: “Air pollution and public health: A PRISMA-compliant systematic review” (https://doi.org/10.3390/atmos8100183); “UK air pollution and public health” (https://doi.org/10.1016/s0140-6736(17)31271-0)]
added information about recent studies l. 340-358
proposed papers were not related to the effect of PAHs exposure
Reviewer 4 Report
Dear Author,
This paper is a review of data collected during “Teplice Program” which monitored airborne pollution, biomarkers and personal exposure over a period of few years (1990-1994) of the population in the district of Teplice and Prachatice, Czech Republic.
The publication Srám RJ, Benes I, Binková B, Dejmek J, Horstman D, Kotĕsovec F, Otto D, Perreault SD, Rubes J, Selevan SG, Skalík I, Stevens RK, Lewtas J. Teplice program--the impact of air pollution on human health. Environ Health Perspect. 1996 Aug;104 Suppl 4(Suppl 4):699-714. doi: 10.1289/ehp.104-1469669. PMID: 8879999; PMCID: PMC1469669.
includes a comprehensive and accurate description of data received from this program. It is a very valuable publication, containing data and its comprehensive analysis.
I wonder why the data is presented in the form of a review after 25 years from the end of the project?
The review was written by Dr Radim Sram who managed the “Teplice Program” 20 years ago if I understand correctly.
I saw that this paper was dedicated to the memory of Dr Blanka Binkova and Dr Joellen Lewtas who provided inspiration and guidance for this project.
What about the rest of the team members who worked on the project?
The Author writes:
(line 15) We obtain …
(line 266, line 302) We hypothesize …
(line 329) We believe that …
etc.
It's a bit awkward considering mono-authorship.
Author writes “Life expectancy in the district of Teplice in 1988 was 64.9 years for males and 73.9 years for females compared to 68.2 years and 75.4 years, respectively, in the Czech Republic [2].” How is it today?
The publication is interesting, but out of date, and unfortunately it is visible at every stage of the work. The results should be related to contemporary research and data.
21 out of 45 literature references items come from before the year 2000. Again, I recommend update this work to a great extent.
The manuscript does not accomplish an in-depth discussion of the results or conclusions as opposed to papers which were previously published on this topic.
Why in Table 1, not all values are given the uncertainty?
Was Figure 2 used in previous studies?
Figure 3 is illegible. Its quality should be improved.
The quality of the English should be improved.
Kind regards
Author Response
Thank you for your comments.
This paper is a review of data collected during “Teplice Program” which monitored airborne pollution, biomarkers and personal exposure over a period of few years (1990-1994) of the population in the district of Teplice and Prachatice, Czech Republic.
The publication Srám RJ, Benes I, Binková B, Dejmek J, Horstman D, Kotĕsovec F, Otto D, Perreault SD, Rubes J, Selevan SG, Skalík I, Stevens RK, Lewtas J. Teplice program--the impact of air pollution on human health. Environ Health Perspect. 1996 Aug;104 Suppl 4(Suppl 4):699-714. doi: 10.1289/ehp.104-1469669. PMID: 8879999; PMCID: PMC1469669.
includes a comprehensive and accurate description of data received from this program. It is a very valuable publication, containing data and its comprehensive analysis.
I wonder why the data is presented in the form of a review after 25 years from the end of the project?
The best answers are given by Reviewers 2 and 3.
added l. 72-73:
Conclusions from Teplice Program are also relevant today because there are many countries that still rely on burning coal for home heating.
The review was written by Dr Radim Sram who managed the “Teplice Program” 20 years ago if I understand correctly.
I saw that this paper was dedicated to the memory of Dr Blanka Binkova and Dr Joellen Lewtas who provided inspiration and guidance for this project.
What about the rest of the team members who worked on the project?
All team members are mentioned in acknowledgement.
The Author writes:
(line 15) We obtain …
(line 266, line 302) We hypothesize …
(line 329) We believe that …
etc.
It's a bit awkward considering mono-authorship.
All these points were corrected.
Author writes “Life expectancy in the district of Teplice in 1988 was 64.9 years for males and 73.9 years for females compared to 68.2 years and 75.4 years, respectively, in the Czech Republic [2].” How is it today?
added: l.283-284
The publication is interesting, but out of date, and unfortunately it is visible at every stage of the work. The results should be related to contemporary research and data.
added: 340-358
21 out of 45 literature references items come from before the year 2000. Again, I recommend update this work to a great extent.
The manuscript does not accomplish an in-depth discussion of the results or conclusions as opposed to papers which were previously published on this topic.
Why in Table 1, not all values are given the uncertainty?
corrected
Was Figure 2 used in previous studies?
no
Figure 3 is illegible. Its quality should be improved.
The quality of the English should be improved.

Round 2
Reviewer 4 Report
Dear Author,
"The best answers are given by Reviewers 2 and 3.
added l. 72-73:
Conclusions from Teplice Program are also relevant today because there are many countries that still rely on burning coal for home heating."
I honestly admit that this is not a fully satisfactory answer to the question. However, I do not want to drill down further, because it is not fully substantive question.
"Life expectancy in the district of Teplice in 1988 was 64.9 years for males and 73.9 years for 280 females compared to 68.2 years and 75.4 years, respectively, in the Czech Republic [2]. Now is life expectancy in the Czech Republic for males 76 years, for females 82 years, but in the mining districts is for males as well as females 2 years shorter."
Some literature reference and a brief comment should be given here. This is an 8-year average increase in life over 38 years. It's a lot. This requires a comment.
Please correct the English language in the publication and if possible update the paper with some another current references.
Yours faithfully
Author Response
Thank you for your comments.
Life expectancy in the district of Teplice in 1988 was 64.9 years for males and 73.9 years for 280 females compared to 68.2 years and 75.4 years, respectively, in the Czech Republic [2]. Now is life expectancy in the Czech Republic for males 76 years, for females 82 years, but in the mining districts is for males as well as females 2 years shorter."
New text was added:
Now is life expectancy in the Czech Republic for males 76 years, for females 82 years, but in the mining districts it is for males as well as females 2 years shorter [44]. The extension of life expectancy after political changes in 1989 were related to the improvements in the health care system and the shift in lifestyles and disease awareness [45]. Modern health policies included screening programs for certain malignant neoplasms, tobacco/smoking bans in public places and ati/alcohol measures [46]. Declines in cardiovascular mortality were associated with improvements in prevention and/or treatment of ischaemic heart disease [47].
English was checked by a native American.

This manuscript is a resubmission of an earlier submission. The following is a list of the peer review reports and author responses from that submission.
Round 1
Reviewer 1 Report
This paper is a review of data accumulated in a study called the “Teplice Program” where exposure to airborne pollution, and various biomarkers and health outcomes, were monitored over a period of about 20 years in Teplice, a region of the Czech Republic with high pollution levels, and Prachatice, a much cleaner region of the same country. This study was an unprecedented pioneering collaboration between the Czech Academy of Sciences and the US Environmental Protection Agency and resulted in a large body of seminal findings. The published papers from the Teplice Program have provided a roadmap for others in the field of molecular epidemiology.
This review was written by Dr. Sram who organized and managed the Teplice Program and chaired the collaboration that made it possible. I believe his intentions may have stemmed from the idea that many of his original publications might have been overlooked by the environmental community, and a review in IJERPH would reach out to an audience of readers who do not read cancer/toxicology journals.
The major drawback of this paper is the quality of the English. I am not aware of the IJERPH policy on language use, but I believe that most journals would require substantial rewriting of this text by an editor or someone who is a native English speaker.
Finally, because this is a review it is to be expected that all of the data are published previously. If this is not the case the author should make clear in the text what data is being published for the first time. In addition, it would be good if Dr. Sram could indicate to the IJERPH Editors the value of publishing now and in this journal as opposed to reviews he has previously published on this topic.

Reviewer 2 Report
Dear Author,
The manuscript presents important results of the research conducted in the region of the Czech Republic in the 1980s and 1990s. Most of the contents are heavily cited from the corresponding author's previous publications. I agree that these studies of the author's exemplary contribution for decades would be in one review paper. However, the manuscript does not accomplish an in-depth discussion of the results incorporating other research performed either contemporary with the studies, and more importantly, discussion with recent research is missing.
The paper is logically and thematically coherent but is lacking in substantial ways. Major ideas related to the content may be ignored or inadequately explored. Ideas and concepts are generally satisfactorily presented although lapses in logic and organization are apparent. Overall, the content and organization need significant revision to represent a critical analysis of the topic.
There are many problems in the English language, including typos, incorrect use of punctuation, units such as micrograms are not in Greek, and missing verbs. Frequent errors in spelling, grammar (such as subject/verb agreements and tense), sentence structure and/or other writing conventions make reading difficult and interfere with comprehensibility. There is some confusion in the proper use of scientific terms. Writing does not flow smoothly from point to point; appropriate transitions are lacking.